# Dual PatchNorm

**Manoj Kumar**                                                                 *mechcoder@google.com*

**Mostafa Dehghani**                                                            *dehghani@google.com*

**Neil Houlsby**                                                                *neilhoulsby@google.com*

*Google Research, Brain Team*

**Reviewed on OpenReview:** *https://openreview.net/forum?id=jgMqve6Qhw*

## Abstract

We propose Dual PatchNorm: two Layer Normalization layers (LayerNorms), before and after the patch embedding layer in Vision Transformers. We demonstrate that Dual Patch-Norm outperforms the result of exhaustive search for alternative LayerNorm placement strategies in the Transformer block itself. In our experiments on image classification, contrastive learning, semantic segmentation and transfer on downstream classification datasets, incorporating this trivial modification, often leads to improved accuracy over well-tuned vanilla Vision Transformers and never hurts.

## 1 Introduction

Layer Normalization (Ba et al., 2016) is key to Transformer's success in achieving both stable training and high performance across a range of tasks. Such normalization is also crucial in Vision Transformers (ViT) (Dosovitskiy et al., 2020; Touvron et al., 2021) which closely follow the standard recipe of the original Transformer model.

Following the "pre-LN" strategy in Baevski & Auli (2019) and Xiong et al. (2020), ViTs place LayerNorms before the self-attention layer and MLP layer in each Transformer block. We explore the following question: Can we improve ViT models with a different LayerNorm ordering? First, across five ViT architectures on ImageNet-1k (Russakovsky et al., 2015), we demonstrate that an exhaustive search of LayerNorm placements between the components of a Transformer block does not improve classification accuracy. This indicates that the pre-LN strategy in ViT is close to optimal. Our observation also applies to other alternate LayerNorm placements: NormFormer (Shleifer et al., 2021) and Sub-LN (Wang et al., 2022), which in isolation, do not improve over strong ViT classification models.

Second, we make an intriguing observation: placing additional LayerNorms before and after the standard ViT-projection layer, which we call Dual PatchNorm (DPN), can improve significantly over well tuned vanilla ViT baselines. Our experiments on image classification across three different datasets with varying number of examples and contrastive learning, demonstrate the efficacy of DPN. Interestingly, our qualitative experiments show that the LayerNorm scale parameters upweight the pixels at the center and corners of each patch.

```
1  hp, wp = patch_size[0], patch_size[1]
2  x = einops.rearrange(
3      x, "b (ht hp) (wt wp) c -> b (ht wt) (hp wp c)", hp=hp, wp=wp)
4  x = nn.LayerNorm(name="ln0")(x)
5  x = nn.Dense(output_features, name="dense")(x)
6  x = nn.LayerNorm(name="ln1")(x)
```

Dual PatchNorm consists of a 2 line change to the standard ViT-projection layer.

## 2    Related Work

Kim et al. (2021) add a LayerNorm after the patch-embedding and show that this improves the robustness of ViT against corruptions on small-scale datasets. Xiao et al. (2021) replace the standard Transformer stem with a small number of stacked stride-two $3 \times 3$ convolutions with batch normalizations and show that this improves the sensitivity to optimization hyperparameters and final accuracy. Xu et al. (2019) analyze LayerNorm and show that the derivatives of mean and variance have a greater contribution to final performance as opposed to forward normalization. Beyer et al. (2022a) consider Image-LN and Patch-LN as alternative strategies to efficiently train a single model for different patch sizes. Wang et al. (2022) add extra LayerNorms before the final dense projection in the self-attention block and the non-linearity in the MLP block, with a different initialization strategy. Shleifer et al. (2021) propose extra LayerNorms after the final dense projection in the self-attention block instead with a LayerNorm after the non-linearity in the MLP block. Unlike previous work, we show that LayerNorms before and after the embedding layer provide consistent improvements on classification and contrastive learning tasks. An orthogonal line of work (Liu et al., 2021; d'Ascoli et al., 2021; Wang et al., 2021) involves incorporating convolutional inductive biases to VisionTransformers. Here, we exclusively and extensively study LayerNorm placements of vanilla ViT.

## 3    Background

### 3.1    Patch Embedding Layer in Vision Transformer

Vision Transformers (Dosovitskiy et al., 2020) consist of a patch embedding layer (PE) followed by a stack of Transformer blocks. The PE layer first rearranges the image $x \in \mathcal{R}^{H \times W \times 3}$ into a sequence of patches $x_p \in \mathcal{R}^{\frac{HW}{P^2} \times P^2}$ where $P$ denotes the patch size. It then projects each patch independently with a dense projection to constitute a sequence of "visual tokens" $\mathbf{x_t} \in \mathcal{R}^{\frac{HW}{P^2} \times D}$ $P$ controls the trade-off between granularity of the visual tokens and the computational cost in the subsequent Transformer layers.

### 3.2    Layer Normalization

Given a sequence of $N$ patches $\mathbf{x} \in \mathcal{R}^{N \times D}$, LayerNorm as applied in ViTs consist of two operations:

$$\mathbf{x} = \frac{\mathbf{x} - \mu(x)}{\sigma(x)} \tag{1}$$

$$\mathbf{y} = \gamma \mathbf{x} + \beta \tag{2}$$

where $\mu(x) \in \mathcal{R}^N, \sigma(x) \in \mathcal{R}^N, \gamma \in \mathcal{R}^D, \beta \in \mathcal{R}^D$.

First, Eq. 1 normalizes each patch $\mathbf{x_i} \in \mathcal{R}^D$ of the sequence to have zero mean and unit standard deviation. Then, Eq 2 applies learnable shifts and scales $\beta$ and $\gamma$ which are shared across all patches.

## 4    Methods

### 4.1    Alternate LayerNorm placements:

Following Baevski & Auli (2019) and Xiong et al. (2020), ViTs incorporate LayerNorm before every self-attention and MLP layer, commonly known as the pre-LN strategy. For each of the self-attention and MLP layer, we evaluate 3 strategies: place LayerNorm before (pre-LN), after (post-LN), before and after (pre+post-LN) leading to nine different combinations.

### 4.2    Dual PatchNorm

Instead of adding LayerNorms to the Transformer block, we also propose to apply LayerNorms in the stem alone, both before and after the patch embedding layer. In particular, we replace

$$\mathbf{x} = \mathrm{PE}(\mathbf{x}) \tag{3}$$

with

$$\mathbf{x} = \mathrm{LN}(\mathrm{PE}(\mathrm{LN}(\mathbf{x}))) \tag{4}$$

and keep the rest of the architecture fixed. We call this Dual PatchNorm (DPN).

## 5 Experiments on ImageNet Classification

### 5.1 Setup

We adopt the standard formulation of Vision Transformers (Sec. 3.1) which has shown broad applicability across a number of vision tasks. We train ViT architectures (with and without DPN) in a supervised fashion on 3 different datasets with varying number of examples: ImageNet-1k (1M), ImageNet-21k (21M) and JFT (4B) (Zhai et al., 2022a). In our experiments, we apply DPN directly on top of the baseline ViT recipes without additional hyperparamter tuning. We split the ImageNet train set into a train and validation split, and use the validation split to arrive at the final DPN recipe.

**ImageNet 1k:** We train 5 architectures: Ti/16, S/16, S/32, B/16 and B/32 using the AugReg (Steiner et al., 2022) recipe for 93000 steps with a batch size of 4096 and report the accuracy on the official ImageNet validation split as is standard practice. The AugReg recipe provides the optimal mixup regularization (Zhang et al., 2017) and RandAugment (Cubuk et al., 2020) for each ViT backbone. Further, we evaluate a S/16 baseline (S/16+) with additional extensive hyperparameter tuning on ImageNet (Beyer et al., 2022b).Finally, we also apply DPN on top of the base and small DeiT variants (Touvron et al., 2021). Our full set of hyperparameters are available in Appendix C and Appendix D.

**ImageNet 21k:** We adopt a similar setup as in ImageNet 1k. We report ImageNet 25 shot accuracies in two training regimes: 93K and 930K steps.

**JFT:** We evaluate the ImageNet 25 shot accuracies of 3 variants (B/32, B/16 and L/16) on 2 training regimes: (220K and 1.1M steps) with a batch size of 4096. In this setup, we do not use any additional data augmentation or mixup regularization.

On ImageNet-1k, we report the 95% confidence interval across atleast 3 independent runs. On ImageNet-21k and JFT, because of expensive training runs, we train each model once and report the mean 25 shot accuracy with 95% confidence interval across 3 random seeds.

### 5.2 DPN versus alternate LayerNorm placements

Each Transformer block in ViT consists of a self-attention (SA) and MLP layer. Following the pre-LN strategy (Xiong et al., 2020), LN is inserted before both the SA and MLP layers. We first show that the default pre-LN strategy in ViT models is close to optimal by evaluating alternate LN placements on ImageNet-1k. We then contrast this with the performance of NormFormer, Sub-LN and DPN.

For each SA and MLP layer, we evaluate three LN placements: Pre, Post and Pre+Post, that leads to nine total LN placement configurations. Additionally, we evaluate the LayerNorm placements in NormFormer (Shleifer et al., 2021) and Sub LayerNorm (Wang et al., 2022) which add additional LayerNorms within each of the self-attention and MLP layers in the transformer block. Figure 1 shows that none of the placements outperform the default Pre-LN strategy significantly, indicating that the default pre-LN strategy is close to optimal. NormFormer provides some improvements on ViT models with a patch size of 32. DPN on the other-hand provides consistent improvements across all 5 architectures.

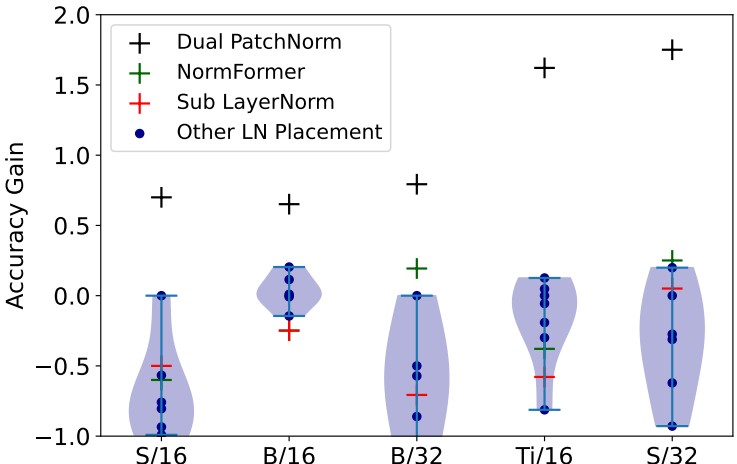

Figure 1: The plot displays the accuracy gains of different LayerNorm placement strategies over the default pre-LN strategy. Each blue point (**Other LN placement**) corresponds to a different LN placement in the Transformer block. None of the placements outperform the default Pre-LN strategy on ImageNet-1k (Russakovsky et al., 2015). Applying DPN (black cross) provides consistent improvements across all 5 architectures.

| Arch | Base | DPN |
|------|------|-----|
| ViT AugReg | | |
| S/32 | $72.1 \pm 0.07$ | $\mathbf{74.0} \pm 0.09$ |
| Ti/16 | $72.5 \pm 0.07$ | $\mathbf{73.9} \pm 0.09$ |
| B/32 | $74.8 \pm 0.06$ | $\mathbf{76.2} \pm 0.07$ |
| S/16 | $78.6 \pm 0.32$ | $\mathbf{79.7} \pm 0.2$ |
| S/16+ | $79.7 \pm 0.09$ | $\mathbf{80.2} \pm 0.03$ |
| B/16 | $80.4 \pm 0.06$ | $\mathbf{81.1} \pm 0.09$ |
| DeiT | | |
| S/16 | $80.1 \pm 0.03$ | $\mathbf{80.4} \pm 0.06$ |
| B/16 | $81.8 \pm 0.03$ | $\mathbf{82.0} \pm 0.05$ |
| AugReg + 384 × 384 Finetune | | |
| B/32 | $79.0 \pm 0.00$ | $\mathbf{80.0} \pm 0.03$ |
| B/16 | $82.2 \pm 0.03$ | $\mathbf{82.8} \pm 0.00$ |

| Arch | Base | DPN |
|------|------|-----|
| 93K Steps | | |
| Ti/16 | $52.2 \pm 0.07$ | $\mathbf{53.6} \pm 0.07$ |
| S/32 | $54.1 \pm 0.03$ | $\mathbf{56.7} \pm 0.03$ |
| B/32 | $60.9 \pm 0.03$ | $\mathbf{63.7} \pm 0.03$ |
| S/16 | $64.3 \pm 0.15$ | $\mathbf{65.0} \pm 0.06$ |
| B/16 | $70.8 \pm 0.09$ | $\mathbf{72.0} \pm 0.03$ |
| 930K Steps | | |
| Ti/16 | $\mathbf{61.0} \pm 0.03$ | $\mathbf{61.2} \pm 0.03$ |
| S/32 | $63.8 \pm 0.00$ | $\mathbf{65.1} \pm 0.12$ |
| B/32 | $72.8 \pm 0.03$ | $\mathbf{73.1} \pm 0.07$ |
| S/16 | $\mathbf{72.5} \pm 0.1$ | $\mathbf{72.5} \pm 0.1$ |
| B/16 | $78.0 \pm 0.06$ | $\mathbf{78.4} \pm 0.03$ |

Table 1: **Left:** ImageNet-1k validation accuracies of five ViT architectures with and without dual patch norm after 93000 steps. **Right:** We train ViT models on ImageNet-21k in two training regimes: 93k and 930k steps with a batch size of 4096. The table shows their ImageNet 25 shot accuracies with and without Dual PatchNorm

## 5.3 Comparison to ViT

In Table 1 left, DPN improved the accuracy of B/16, the best ViT model by 0.7 while S/32 obtains the maximum accuracy gain of 1.9. The average gain across all architecture is 1.4. On top of DeiT-S and DeiT-B, DPN provides an improvement of 0.3 and 0.2 respectively. Further, we finetune B/16 and B/32 models with and without DPN on high resolution ImageNet (384 × 384) for 5000 steps with a batch-size of 512 (See Appendix D for the full hyperparameter setting). Applying DPN improves high-res, finetuned B/16 and B/32 by 0.6 and 1.0 respectively.

DPN improves all architectures trained on ImageNet-21k (Table 1 Right) and JFT (Table 2) on shorter training regimes with average gains of 1.7 and 0.8 respectively. On longer training regimes, DPN improves the accuracy of the best-performing architectures on JFT and ImageNet-21k by 0.5 and 0.4 respectively.

In three cases, Ti/16 and S/32 with ImageNet-21k and B/16 with JFT, DPN matches or leads to marginally worse results than the baseline. Nevertheless, across a large fraction of ViT models, simply employing DPN out-of-the-box on top of well-tuned ViT baselines lead to significant improvements.

### 5.4 Finetuning on ImageNet with DPN

We finetune four models trained on JFT-4B with two resolutions on ImageNet-1k: (B/32, B/16) × (220K, 1.1M) steps on resolutions $224 \times 224$ and $384 \times 384$. On B/32 we observe a consistent improvement across all configurations. With L/16, DPN outperforms the baseline on 3 out of 4 configurations.

| Arch | Base | DPN |
|------|------|-----|
| | 220K steps | |
| B/32 | $63.8_{\pm 0.03}$ | $\mathbf{65.2}_{\pm 0.03}$ |
| B/16 | $72.1_{\pm 0.09}$ | $\mathbf{72.4}_{\pm 0.07}$ |
| L/16 | $77.3_{\pm 0.00}$ | $\mathbf{77.9}_{\pm 0.06}$ |
| | 1.1M steps | |
| B/32 | $70.7_{\pm 0.1}$ | $\mathbf{71.1}_{\pm 0.09}$ |
| B/16 | $\mathbf{76.9}_{\pm 0.03}$ | $76.6_{\pm 0.03}$ |
| L/16 | $80.9_{\pm 0.03}$ | $\mathbf{81.4}_{\pm 0.06}$ |

| Arch | Resolution | Steps | Base | DPN |
|------|-----------|-------|------|-----|
| B/32 | 224 | 220K | $77.6_{\pm 0.06}$ | $\mathbf{78.3}_{\pm 0.00}$ |
| B/32 | 384 | 220K | $81.3_{\pm 0.09}$ | $\mathbf{81.6}_{\pm 0.00}$ |
| B/32 | 224 | 1.1M | $80.8_{\pm 0.1}$ | $\mathbf{81.3}_{\pm 0.00}$ |
| B/32 | 384 | 1.1M | $83.8_{\pm 0.03}$ | $\mathbf{84.1}_{\pm 0.00}$ |
| L/16 | 224 | 220K | $84.9_{\pm 0.06}$ | $\mathbf{85.3}_{\pm 0.03}$ |
| L/16 | 384 | 220K | $86.7_{\pm 0.03}$ | $\mathbf{87.0}_{\pm 0.00}$ |
| L/16 | 224 | 1.1M | $86.7_{\pm 0.03}$ | $\mathbf{87.1}_{\pm 0.00}$ |
| L/16 | 384 | 1.1M | $\mathbf{88.2}_{\pm 0.00}$ | $\mathbf{88.3}_{\pm 0.06}$ |

Table 2: **Left:** We train 3 ViT models on JFT-4B in two training regimes: 200K and 1.1M steps with a batch size of 4096. The table displays their ImageNet 25 shot accuracies with and without DPN. **Right:** Corresponding full finetuneing results on ImageNet-1k.

## 6 Experiments on Downstream Tasks

### 6.1 Finetuning on VTAB

We finetune ImageNet-pretrained B/16 and B/32 with and without DPN on the Visual Task Adaption benchmark (VTAB) (Zhai et al., 2019). VTAB consists of 19 datasets: 7 ● Natural , 4 ● Specialized and 8 ● Structured . ● Natural consist of datasets with natural images captured with standard cameras, ● Specialized has images captured with specialized equipment and ● Structured require scene comprehension. We use the VTAB training protocol which defines a standard train split of 800 examples and a validation split of 200 examples per dataset. We perform a lightweight sweep across 3 learning rates on each dataset and use the mean validation accuracy across 3 seeds to pick the best model. Appendix E references the standard VTAB finetuning configuration. We then report the corresponding mean test score across 3 seeds in Table 3. In Table 3, accuracies within 95% confidence interval are not bolded.

On ● Natural , which has datasets closest to the source dataset ImageNet, B/32 and B/16 with DPN significantly outperform the baseline on 7 out of 7 and 6 out of 7 datasets respectively. Sun397 (Xiao et al., 2010) is the only dataset where applying DPN performs worse. In Appendix F, we additionally show that DPN helps when B/16 is trained from scratch on Sun397. Applying DPN on ● Structured improves accuracy on 4 out of 8 datasets and remains neutral on 2 on both B/16 and B/32. On ● Specialized , DPN improves on 1 out of 4 datasets, and is neutral on 2. To conclude, DPN offers the biggest improvements, when finetuned on ● Natural . On ● Structured and ● Specialized , DPN is a lightweight alternative, that can help or at least not hurt on a majority of datasets.

| | Caltech101 | CIFAR-100 | DTD | Flowers102 | Pets | Sun397 | SVHN | Camelyon | EuroSAT | Resisc45 | Retinopathy |
|---|---|---|---|---|---|---|---|---|---|---|---|
| B/32 | 87.1 | 53.7 | 56.0 | 83.9 | 87.2 | 32.0 | 76.8 | 77.9 | 94.8 | 78.2 | **71.2** |
| + DPN | **87.7** | **58.1** | **60.7** | **86.4** | **88.0** | **35.4** | **80.3** | 78.5 | 95.0 | **81.6** | 70.3 |
| B/16 | 86.1 | 35.5 | 60.1 | 90.8 | 90.9 | **33.9** | 76.7 | 81.3 | 95.9 | 81.2 | **74.7** |
| + DPN | **86.6** | **51.4** | **63.1** | **91.3** | **92.1** | 32.5 | **78.3** | 80.6 | 95.8 | **83.5** | 73.3 |

| | Clevr-Count | Clevr-Dist | DMLab | dSpr-Loc | dSpr-Ori | KITTI-Dist | sNORB-Azim | sNORB-Elev |
|---|---|---|---|---|---|---|---|---|
| B/32 | 58.3 | 52.6 | 39.2 | **71.3** | 59.8 | 73.6 | 20.7 | **47.2** |
| + DPN | **62.5** | **55.5** | **40.7** | 60.8 | **61.6** | 73.4 | 20.9 | 34.4 |
| B/16 | 65.2 | **59.8** | 39.7 | 72.1 | 61.9 | 81.3 | 18.9 | **50.4** |
| + DPN | **73.7** | 48.3 | **41.0** | 72.4 | **63.0** | 80.6 | **21.6** | 36.2 |

Table 3: We evaluate DPN on VTAB (Zhai et al., 2019). When finetuned on ● Natural , B/32 and B/16 with DPN significantly outperform the baseline on 7 out of 7 and 6 out of 7 datasets respectively. On ● Structured , DPN improves both B/16 and B/32 on 4 out of 8 datasets and remains neutral on 2. On ● Specialized , DPN improves on 1 out of 4 datasets, and is neutral on 2.

## 6.2 Contrastive Learning

We apply DPN on image-text contrastive learning (Radford et al., 2021). Each minibatch consists of a set of image and text pairs. We train a text and image encoder to map an image to its correct text over all other texts in a minibatch. Specifically, we adopt LiT (Zhai et al., 2022b), where we initialize and freeze the image encoder from a pretrained checkpoint and train the text encoder from scratch. To evaluate zero-shot ImageNet accuracy, we represent each ImageNet class by its text label, which the text encoder maps into a class embedding. For a given image embedding, the prediction is the class corresponding to the nearest class embedding.

We evalute 4 frozen image encoders: 2 architectures (B/32 and L/16) trained with 2 schedules (220K and 1.1M steps). We resue standard hyperparameters and train only the text encoder using a contrastive loss for 55000 steps with a batch-size of 16384. Table 4 shows that on B/32, DPN improves over the baselines on both the setups while on L/16 DPN provides improvement when the image encoder is trained with shorter training schedules.

## 6.3 Semantic Segmentation

We finetune ImageNet-pretrained B/16 with and without DPN on the ADE-20K $512 \times 512$ (Zhou et al., 2019) semantic segmentation task. Following Strudel et al. (2021), a single dense layer maps the ViT features into per-patch output logits. A bilinear upsampling layer then transforms the output distribution into the final high resolution $512 \times 512$ semantic segmentation output. We finetune the entire ViT backbone with standard

| Arch | Steps | Base | DPN |
|------|-------|------|-----|
| B/32 | 220K | $61.9_{\pm 0.12}$ | $\mathbf{63.0}_{\pm 0.09}$ |
| B/32 | 1.1M | $67.4_{\pm 0.07}$ | $\mathbf{68.0}_{\pm 0.09}$ |
| L/16 | 220K | $75.0_{\pm 0.11}$ | $\mathbf{75.4}_{\pm 0.00}$ |
| L/16 | 1.1M | $\mathbf{78.7}_{\pm 0.05}$ | $\mathbf{78.7}_{\pm 0.1}$ |

Table 4: Zero Shot ImageNet accuracy on the LiT (Zhai et al., 2022b) contrastive learning setup.

| Fraction of Train Data | 1/16 | 1/8 | 1/4 | 1/2 | 1 |
|------------------------|------|-----|-----|-----|---|
| B/16 | $27.3_{\pm 0.09}$ | $32.6_{\pm 0.09}$ | $36.9_{\pm 0.13}$ | $40.8_{\pm 0.1}$ | $45.6_{\pm 0.08}$ |
| +DPN | $\mathbf{28.0}_{\pm 0.21}$ | $\mathbf{33.7}_{\pm 0.11}$ | $\mathbf{38.0}_{\pm 0.11}$ | $\mathbf{41.9}_{\pm 0.09}$ | $\mathbf{46.1}_{\pm 0.11}$ |

Table 5: We finetune ImageNet pretrained B/16 models with and without DPN on the ADE20K Semantic Segmentation task, when a varying fraction of ADE20K training data is available. The table reports the mean IoU across ten random seeds. Applying DPN improves IoU across all settings.

per-pixel cross-entropy loss. Appendix G specifies the full set of finetuning hyperparameters. Table 5 reports the mean mIOU across 10 random seeds and on different fractions of training data. The improvement in IoU is consistent across all setups.

# 7 Ablations

**Is normalizing both the inputs and outputs of the embedding layer optimal?** In Eq 4, DPN applies LN to both the inputs and outputs to the embedding layer. We assess three alternate strategies: **Pre**, **Post** and **Post PosEmb** (Radford et al., 2021). **Pre** applies LayerNorm only to the inputs, **Post** only to the outputs and **Post PosEmb** to the outputs after being summed with positional embeddings.

Table 6 displays the accuracy gains with two alternate strategies: **Pre** is unstable on B/32 leading to a significant drop in accuracy. Additionally, **Pre** obtains minor drops in accuracy on S/32 and Ti/16. **Post** and **Post PosEmb** achieve worse performance on smaller models B/32, S/32 and Ti/16. Our experiments show that applying LayerNorm to both inputs and outputs of the embedding layer is necessary to obtain consistent improvements in accuracy across all ViT variants.

|  | B/16 | S/16 | B/32 | S/32 | Ti/16 |
|--|------|------|------|------|-------|
| Pre | -0.1 | 0.0 | -2.6 | -0.2 | -0.3 |
| Post | 0.0 | -0.2 | -0.5 | -0.7 | -1.1 |
| Post PosEmb | 0.0 | -0.1 | -0.4 | -0.9 | -1.1 |
| Only learnable | -0.8 | -0.9 | -1.2 | -1.6 | -1.6 |
| RMSNorm | 0.0 | -0.1 | -0.4 | -0.5 | -1.7 |
| No learnable | -0.5 | 0.0 | -0.2 | -0.1 | -0.1 |

Table 6: Ablations of various components of DPN. **Pre:** LayerNorm only to the inputs of the embedding layer. **Post:** LayerNorm only to the outputs of the embedding layer. **No learnable:** Per-patch normalization without learnable LayerNorm parameters. **Only learnable:** Learnable scales and shifts without standardization.

**Normalization vs Learnable Parameters:** As seen in Sec. 3.2, LayerNorm constitutes a normalization operation followed by learnable scales and shifts. We also ablate the effect of each of these operations in DPN.

Applying only learnable scales and shifts without normalization leads to a significant decrease in accuracy across all architectures. (See: **Only learnable** in Table 6). Additionally, removing the learnable parameters leads to unstable training on B/16 (**No learnable** in Table 6). Finally, removing the centering and bias parameters as done in **RMSNorm** (Zhang & Sennrich, 2019), reduces the accuracy of B/32, S/32 and Ti/16. We conclude that while both normalization and learnable parameters contribute to the success of DPN, normalization has a higher impact.

## 8 Analysis

### 8.1 Gradient Norm Scale

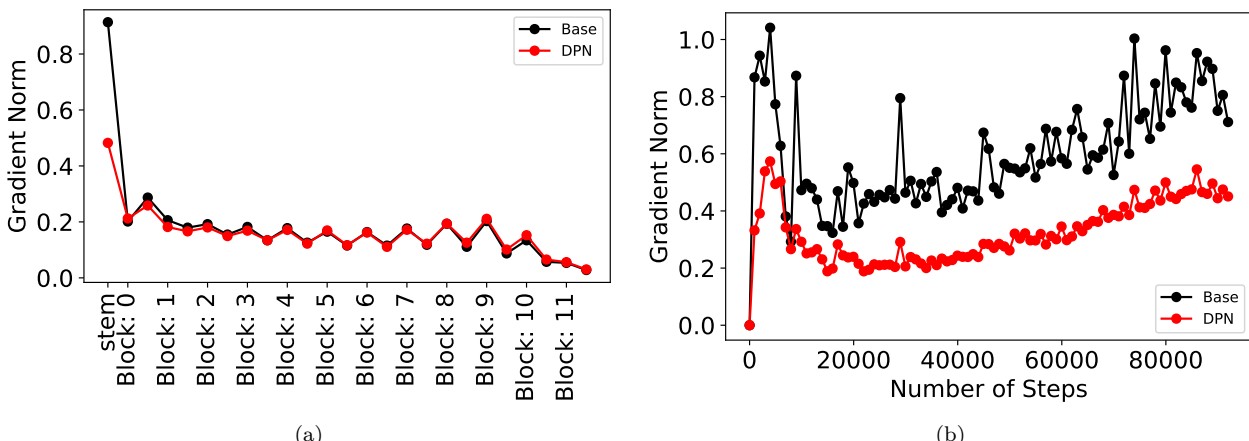

(a)                                                    (b)

Figure 2: Gradient Norms with and without DPN in B/16. **Left:** Gradient Norm vs Depth. **Right:** Gradient Norm of the embedding layer vs number of steps.

We report per-layer gradient norms with and without DPN on B/16. Fig. 2 (Left) plots the mean gradient norm of the last 1000 training steps as a function of depth with and without DPN. Interestingly, the gradient norm of the base ViT patch embedding (black) is disproportionately large compared to the other layers. Applying DPN (red), on the other hand, scales down the gradient norm of the embedding layer. Fig. 2 (Right) additionally shows that the gradient norm of the embedding layer is reduced not only before convergence but also throughout the course of training. This property is consistent across ViT architectures of different sizes (Appendix H).

### 8.2 Visualizing Scale Parameters

Note that the first LayerNorm in Eq. 4 is applied directly on patches, that is, to raw pixels. Thus, the learnable parameters (biases and scales) of the first LayerNorm can be visualized directly in pixel space. Fig. 3 shows the scales of our smallest model and largest model which are: Ti/16 trained on ImageNet for 90000 steps and L/16 trained on JFT for 1.1M steps respectively. Since the absolute magnitude of the scale parameters vary across the R, G and B channel, we visualize the scale separately for each channel. Interestingly, for both models the scale parameter increases the weight of the pixels in the center of the patch and at the corners.

## 9 Conclusion

We propose a simple modification to vanilla ViT models and show its efficacy on classification, contrastive learning, semantic segmentation and transfer to small classification datasets.

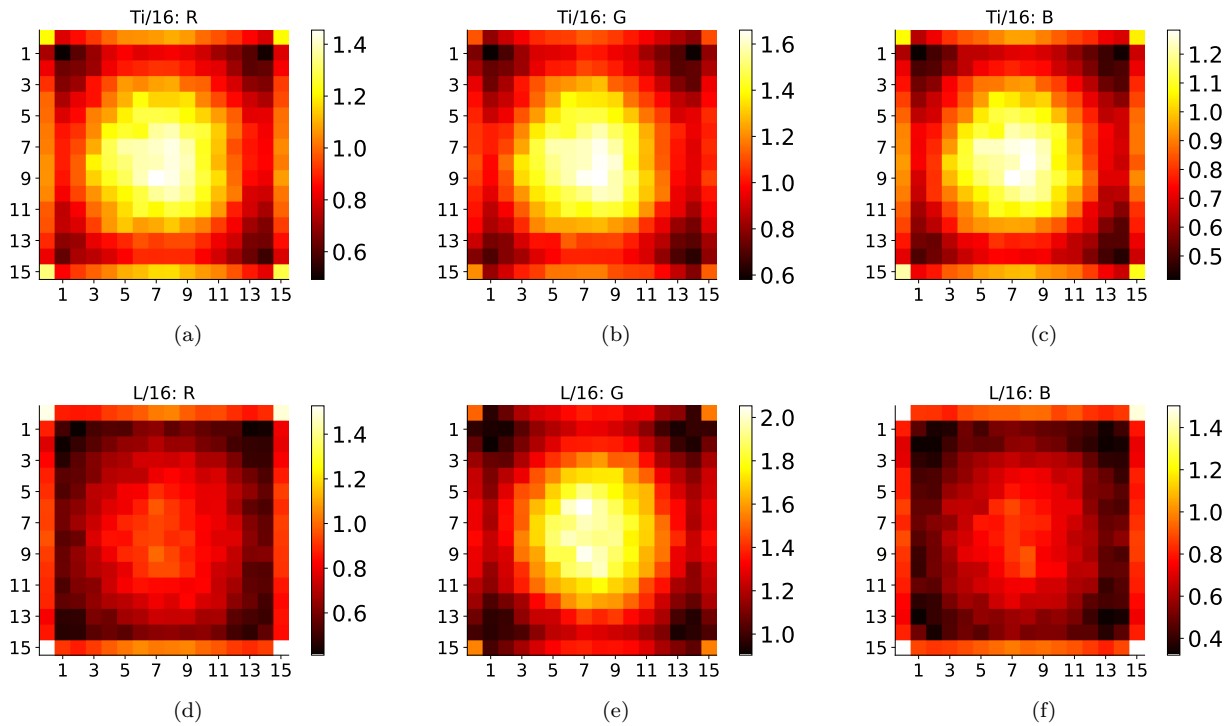

Figure 3: Visualization of scale parameters of the first LayerNorm. **Top:** Ti/16 trained on ImageNet 1k. **Bottom:** L/16 trained on JFT-4B

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

## A    Initial Project Idea

We arrived at the Dual PatchNorm solution because of another project that explored adding whitened (decorrelated) patches to ViT. Our initial prototype had a LayerNorm right after the decorrelated patches, to ensure that they are of an appropriate scale. This lead to improvements across multiple benchmarks, suggesting that whitened patches can improve image classification. We later found out via ablations, that just LayerNorm is sufficient at the inputs and adding whitened patches on their own could degrade performance. Our paper highlights the need for rigorous ablations of complicated algorithms to arrive at simpler solutions which can be equally or even more effective.

## B    Code

We perform all our experiments in the big-vision (Beyer et al., 2022c) and Scenic (Dehghani et al., 2022) library. Since the first LayerNorm of DPN is directly applied on pixels, we replace the first convolution with a patchify operation implemented with the einops (Rogozhnikov, 2022) library and a dense projection.

## C    ViT AugReg: Training Configurations

```
1  import big_vision.configs.common as bvcc
2  from big_vision.configs.common_fewshot import get_fewshot_lsr
3  import ml_collections as mlc
4
5
6  RANDAUG_DEF = {
7      'none': '',
8      'light1': 'randaug(2,0)',
9      'light2': 'randaug(2,10)',
10     'medium1': 'randaug(2,15)',
11     'medium2': 'randaug(2,15)',
12     'strong1': 'randaug(2,20)',
13     'strong2': 'randaug(2,20)',
14 }
15
16 MIXUP_DEF = {
17     'none': dict(p=0.0, fold_in=None),
18     'light1': dict(p=0.0, fold_in=None),
19     'light2': dict(p=0.2, fold_in=None),
```

```
20      'medium1': dict(p=0.2, fold_in=None),
21      'medium2': dict(p=0.5, fold_in=None),
22      'strong1': dict(p=0.5, fold_in=None),
23      'strong2': dict(p=0.8, fold_in=None),
24  }
25
26
27  def get_config(arg=None):
28      """Config for training."""
29      arg = bvcc.parse_arg(arg, variant='B/32', runlocal=False, aug='')
30      config = mlc.ConfigDict()
31
32      config.pp_modules = ['ops_general', 'ops_image']
33      config.init_head_bias = -6.9
34      variant = 'B/16'
35
36      aug_setting = arg.aug or {
37          'Ti/16': 'light1',
38          'S/32': 'medium1',
39          'S/16': 'medium2',
40          'B/32': 'medium2',
41          'B/16': 'medium2',
42          'L/16': 'medium2',
43      }[variant]
44
45      config.input = dict()
46      config.input.data = dict(
47          name='imagenet2012',
48          split='train[:99%]',
49      )
50      config.input.batch_size = 4096
51      config.input.cache_raw = True
52      config.input.shuffle_buffer_size = 250_000
53
54      pp_common = (
55          '|value_range(-1, 1)'
56          '|onehot(1000, key="{lbl}", key_result="labels")'
57          '|keep("image", "labels")'
58      )
59
60      config.input.pp = (
61          'decode_jpeg_and_inception_crop(224)|flip_lr|' +
62          RANDAUG_DEF[aug_setting] +
63          pp_common.format(lbl='label')
64      )
65      pp_eval = 'decode|resize_small(256)|central_crop(224)' + pp_common
66      config.input.prefetch = 8
67
68      config.num_classes = 1000
69      config.loss = 'sigmoid_xent'
70      config.total_epochs = 300
71      config.log_training_steps = 50
72      config.ckpt_steps = 1000
73
74      # Model section
75      config.model_name = 'vit'
76      config.model = dict(
77          variant=variant,
78          rep_size=True,
79          pool_type='tok',
80          dropout=0.1,
81          stoch_depth=0.1,
82          stem_ln='dpn')
83
84      # Optimizer section
85      config.grad_clip_norm = 1.0
86      config.optax_name = 'scale_by_adam'
87      config.optax = dict(mu_dtype='bfloat16')
```

```
88
89    config.lr = 0.001
90    config.wd = 0.0001
91    config.seed = 0
92    config.schedule = dict(warmup_steps=10_000, decay_type='cosine')
93
94    config.mixup = MIXUP_DEF[aug_setting]
95
96    # Eval section
97    def get_eval(split, dataset='imagenet2012'):
98      return dict(
99          type='classification',
100         data=dict(name=dataset, split=split),
101         pp_fn=pp_eval.format(lbl='label'),
102         loss_name=config.loss,
103         log_steps=2500,
104         cache_final=not arg.runlocal,
105     )
106   config.evals = {}
107   config.evals.train = get_eval('train[:2%]')
108   config.evals.minival = get_eval('train[99%:]')
109   config.evals.val = get_eval('validation')
110   return config
```

AugReg Recipe: B/16.

For smaller models (S/32, Ti/16 and S/16), as per the AugReg recipe, we switch off stochastic depth and dropout. For S/32, we also set representation size to be false.

## D   ViT AugReg: High Res Finetuning

```
1  import ml_collections as mlc
2
3
4  def get_config(runlocal=False):
5    """Config for adaptation on imagenet."""
6    config = mlc.ConfigDict()
7
8    config.loss = 'sigmoid_xent'
9    config.num_classes = 1000
10   config.total_steps = 5000
11   config.pp_modules = ['ops_general', 'ops_image']
12
13   config.seed = 0
14   config.input = {}
15   config.input.data = dict(
16       name='imagenet2012',
17       split='train[:99%]',
18   )
19   config.input.batch_size = 512 if not runlocal else 8
20   config.input.shuffle_buffer_size = 50_000 if not runlocal else 100
21   config.input.cache_raw = True
22   variant = 'B/32'
23
24   pp_common = (
25       'value_range(-1, 1)|'
26       'onehot(1000, key="{lbl}", key_result="labels")|'
27       'keep("image", "labels")'
28   )
29   config.input.pp = (
30       'decode_jpeg_and_inception_crop(384)|flip_lr|' +
31       pp_common.format(lbl='label')
32   )
33   pp_eval = 'decode|resize_small(418)|central_crop(384)|' + pp_common
34
35   config.log_training_steps = 10
36   config.ckpt_steps = 1000
```

```
37
38   config.model_name = 'vit'
39   config.model_init = 'low_res/path'
40   config.model = dict(variant=variant, pool_type='tok', stem_ln='dpn', rep_size=True)
41
42   config.model_load = dict(dont_load=['head/kernel', 'head/bias'])
43
44   # Optimizer section
45   config.optax_name = 'big_vision.momentum_hp'
46   config.grad_clip_norm = 1.0
47   config.wd = None
48   config.lr = 0.03
49   config.schedule = dict(
50       warmup_steps=500,
51       decay_type='cosine',
52   )
53
54   # Eval section
55   def get_eval(split, dataset='imagenet2012'):
56     return dict(
57         type='classification',
58         data=dict(name=dataset, split=split),
59         pp_fn=pp_eval.format(lbl='label'),
60         loss_name=config.loss,
61         log_steps=2500,
62         cache_final=not runlocal,
63     )
64   config.evals = {}
65   config.evals.train = get_eval('train[:2%]')
66   config.evals.minival = get_eval('train[99%:]')
67   config.evals.val = get_eval('validation')
68
69   return config
```

High Resolution Finetuning

## E   VTAB Finetuneing

```
1  from ml_collections import ConfigDict
2
3
4  def get_config():
5    """Config for adaptation on VTAB."""
6    config = ConfigDict()
7
8    config.loss = 'sigmoid_xent'
9    config.num_classes = 0
10   config.total_steps = 2500
11   config.pp_modules = ['ops_general', 'ops_image', 'proj.vtab.pp_ops']
12
13   config.seed = 0
14   config.input = dict()
15   config.input.data = dict(
16       name='',
17       split='train[:800]',
18   )
19   config.input.batch_size = 512
20   config.input.shuffle_buffer_size = 50_000
21   config.input.cache_raw = False
22
23   config.input.pp = ''
24   config.log_training_steps = 10
25   config.log_eval_steps = 100
26   config.ckpt_steps = 1000
27   config.ckpt_timeout = 1
28
29   config.prefetch_to_device = 2
```

```
30
31   # Model.
32   config.model_name = 'vit'
33   stem_ln = 'dpn'
34   variant = 'B/32'
35
36   config.model_init = model_inits[variant][stem_ln]
37   config.model = dict(
38       variant=variant,
39       rep_size=True,
40       pool_type='tok',
41       stem_ln=stem_ln)
42   config.model_load = dict(dont_load=['head/kernel', 'head/bias'])
43
44   # Optimizer section
45   config.optax_name = 'big_vision.momentum_hp'
46   config.grad_clip_norm = 1.0
47   config.wd = None
48   config.lr = 0.0003
49   config.ckpt_timeout = 3600
50   config.schedule = dict(
51       warmup_steps=200,
52       decay_type='cosine',
53   )
54
55   return config
```

High Resolution Finetuning

## F  SUN397: Train from scratch

On Sun397, applying DPN improves ViT models trained from scratch. We first search for an optimal hyperparameter setting across 3 learning rates: 1e-3, 3e-4, 1e-4, 2 weight decays: 0.03, 0.1 and two dropout values: 0.0, 0.1. We then searched across 3 mixup values 0.0, 0.2 and 0.5 and 4 randaugment distortion magnitudes 0, 5, 10 and 15. We train the final config for 600 epochs.

|                | Base | DPN  |
| -------------- | ---- | ---- |
|                | 41.4 | **47.5** |
| + Augmentation | 48.3 | **50.7** |
| + Train Longer | 52.5 | **56.0** |

|                | Base | DPN  |
| -------------- | ---- | ---- |
|                | 45.6 | **51.8** |
| + Augmentation | 58.7 | **63.0** |
| + Train Longer | 60.8 | **66.3** |

Table 7: Sun train from scratch. **Left:** B/32 and **Right:** B/16

## G  Semantic Segmentation Hyperparameter

```
1  def get_config():
2    """Returns the base experiment configuration for Segmentation on ADE20k."""
3    config = ml_collections.ConfigDict()
4    config.experiment_name = 'linear_decoder_semseg_ade20k'
5
6    # Dataset.
7    config.dataset_name = 'semseg_dataset'
8    config.dataset_configs = ml_collections.ConfigDict()
9    config.dataset_configs.name = 'ade20k'
10   config.dataset_configs.use_coarse_training_data = False
11   config.dataset_configs.train_data_pct = 100
12   mean_std = '[0.485, 0.456, 0.406], [0.229, 0.224, 0.225]'
13   common = (
14       '|standardize(' + mean_std + ', data_key="inputs")'
15       '|keep("inputs", "label")')
16   config.dataset_configs.pp_train = (
17       'mmseg_style_resize(img_scale=(2048, 512), ratio_range=(0.5, 2.0))'
```

```
18          '|random_crop_with_mask(size=512, cat_max=0.75, ignore_label=0)'
19          '|flip_with_mask'
20          '|squeeze(data_key="label")'
21          '|photometricdistortion(data_key="inputs")') + common
22      config.dataset_configs.max_size_train = 512
23      config.dataset_configs.pp_eval = (
24          'squeeze(data_key="label")') + common
25      config.dataset_configs.pp_test = (
26        'multiscaleflipaug(data_key="inputs")'
27        '|squeeze(data_key="label")') + common
28
29      # Model.
30      version, patch = VARIANT.split('/')
31      config.model = ml_collections.ConfigDict()
32      config.model.hidden_size = {'Ti': 192,
33                                  'S': 384,
34                                  'B': 768,
35                                  'L': 1024,
36                                  'H': 1280}[version]
37      config.model.patches = ml_collections.ConfigDict()
38      config.model.patches.size = [int(patch), int(patch)]
39      config.model.num_heads = {'Ti': 3, 'S': 6, 'B': 12, 'L': 16, 'H': 16}[version]
40      config.model.mlp_dim = {'Ti': 768,
41                              'S': 1536,
42                              'B': 3072,
43                              'L': 4096,
44                              'H': 5120}[version]
45      config.model.num_layers = {'Ti': 12,
46                                 'S': 12,
47                                 'B': 12,
48                                 'L': 24,
49                                 'H': 32}[version]
50      config.model.attention_dropout_rate = 0.0
51      config.model.dropout_rate = 0.0
52      config.model.dropout_rate_last = 0.0
53      config.model.stochastic_depth = 0.1
54      config.model_dtype_str = 'float32'
55      config.model.pos_interpolation_method = 'bilinear'
56      config.model.pooling = 'tok'
57      config.model.concat_backbone_output = False
58      config.pretrained_path = ''
59      config.pretrained_name = 'dpn_b16'
60      config.model.posembs = (32, 32)  # 512 / 16
61      config.model.positional_embedding = 'learned'
62      config.model.upernet = False
63      config.model.fcn = True
64      config.model.auxiliary_loss = -1
65      config.model.out_with_norm = False
66      config.model.use_batchnorm = False
67      config.model.dpn = True
68
69      # Trainer.
70      config.trainer_name = 'segmentation_trainer'
71      config.eval_only = False
72      config.oracle_eval = False
73      config.window_stride = 341
74
75      # Optimizer.
76      config.optimizer = 'adamw'
77      config.weight_decay = 0.01
78      config.freeze_backbone = False
79      config.layerwise_decay = 0.
80      config.skip_scale_and_bias_regularization = True
81      config.optimizer_configs = ml_collections.ConfigDict()
82
83      config.batch_size = 16
84      config.num_training_epochs = 128
85      config.max_grad_norm = None
```

```
86   config.label_smoothing = None
87   config.class_rebalancing_factor = 0.0
88   config.rng_seed = 0
89
90   # Learning rate.
91   config.steps_per_epoch = 20210 // config.batch_size
92   config.total_steps = config.num_training_epochs * config.steps_per_epoch
93   config.lr_configs = ml_collections.ConfigDict()
94   config.lr_configs.learning_rate_schedule = 'compound'
95   config.lr_configs.factors = 'constant * polynomial * linear_warmup'
96   config.lr_configs.warmup_steps = 0
97   config.lr_configs.decay_steps = config.total_steps
98   config.lr_configs.base_learning_rate = 0.00003
99   config.lr_configs.end_factor = 0.
100  config.lr_configs.power = 0.9
101  return config
```

Semantic Segmentation Config

## H  Gradient Norm Scale

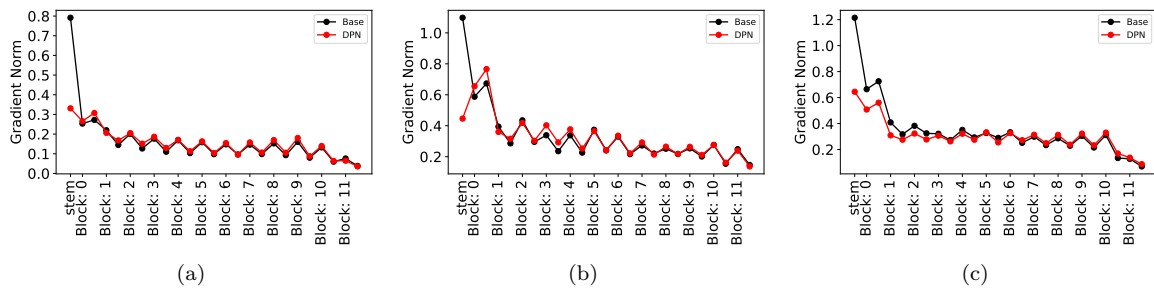

(a)                    (b)                    (c)

Figure 4: Gradient Norm vs Depth. **Left:** B/32. **Center:** S/32 **Right:** S/16

.

