# OpenReview forum: "Dual PatchNorm"
_TMLR — Accepted by TMLR_

### Review · Reviewer_tHeX · 2023-03-06

**Summary Of Contributions:**

The paper proposes a simple modification to the normalization strategy used for the patch embedding layer in Vision Transformers (ViTs).
The proposed recipe, DualPatchNorm or DPN, is simple: Layer Normalization layers (LayerNorm) are applied before and after the patch embedding layer. The authors show that this modification helps in many scenarios for supervised classification on imagenet and contrastive learning

**Audience:**

Yes

**Claims And Evidence:**

Yes

**Requested Changes:**

* **Transfer learning experiments:** Using the learned models for transfer experiments would show if the added robustness/performance gains that the DPN offers persists for generalization to other datasets (eg fine-tuning or after freezing the model and learning classifiers)

* **Tables with hyper parameters**: To be more reproducible, Tables with all model and optimization hyperparameters used for the experiments (on Imagenet1k and 21k) are needed ( e.g. something similar to Tab11 in the latest arxiv version of the MAE paper [He et al 2022])

* Clarify "zero-shot ImageNet accuracy" for the contrastive learning experiments - does this mean 1-NN retrieval? Please add more info on the experimental setup

* The visualization of the scale parameter on the first layer, that is not really explored more deeply - why are the corner patches so important? Why is there such variance across color channels?

* Experiments on other tasks: It would be great to see if this also transfers to ViTs for other tasks, eg object detection or semantic segmentation, where ViT are now used.

[He et al 2022] Masked Autoencoders Are Scalable Vision Learners


**Strengths And Weaknesses:**

Strengths:

* The paper is clearly written and straightforward. It shows pseudocode and reports gains in many cases
* the gains are across different architecture sizes

Weaknesses:

* it only presents results on imagenet (and a non-public propriety dataset) and for testing on the training classes (apart from the non-reproducible JFT finetuning experiment). No transfer experiments are shown, eg to other smaller datasets with classes beyond the ones in the training set, and no experiments for ViTs beyond classification
* This is an empirical study with no effort to try to explore or understand where the gains are coming from (beyond a visualization of the scale parameter on the first layer)

---

> ### Author Response · Authors · 2023-03-23
> **Response**
>
> Thanks for the review! We updated a revision with all changes marked in red.
>
> -------------------------------------------
> ## Transfer learning: Finetuning on VTAB
>
> We finetune ImageNet-pretrained B/16 and B/32 with and without DPN on the Visual Task Adaption benchmark (VTAB). VTAB consists of 19 datasets: 7 Natural , 4 Specialized  and 8 Structured . Natural consist of datasets with natural images captured with standard cameras, Specialized has images captured with specialized equipment and \taskStructured require scene comprehension. We use the VTAB training protocol which defines a standard train split of 800 examples and a validation split of 200 examples per dataset. We perform a lightweight sweep across 3 learning rates on each dataset and use the mean validation accuracy across 3 seeds to pick the best model. Appendix E references the standard VTAB finetuning configuration. We then report the corresponding mean test score across 3 seeds in Table 3. In Table 3 accuracies within $95 \%$ confidence interval are not bolded.}
>
> On Natural, which has datasets closest to the source dataset ImageNet, B/32 and B/16 with DPN significantly outperform the baseline on 7 out of 7 and 6 out of 7 datasets respectively. Sun397 is the only dataset where applying DPN performs worse. In Appendix F, we additionally show that DPN helps when B/16 is trained from scratch on Sun397. Applying DPN on Structured improves accuracy on 4 out of 8 datasets and remains neutral on 2 on both B/16 and B/32. On Specialized, DPN improves on 1 out of 4 datasets, and is neutral on 2. To conclude, DPN offers the biggest improvements, when finetuned on Natural. On Structured and Specialized, DPN is a lightweight alternative, that can help or at least not hurt on a majority of datasets.
>
> ------------
>
> ## Downstream: Semantic Segmentation
>
> We finetune ImageNet-pretrained B/16 with and without DPN on the ADE-20K $512 \times 512$ Semantic Segmentation task. Following [Strudel2021 et. al], a single dense layer maps the ViT features into per-patch output logits. A bilinear upsampling layer then transforms the output distribution into the final high resolution $512 \times 512$ semantic segmentation output. We finetune the entire ViT backbone with standard per-pixel cross-entropy loss. Appendix G specifies the full set of finetuning hyperparameters. Table 5 reports the mean mIOU across 10 random seeds and on different fractions of training data. The improvement in IoU is consistent across all setups.
>
> ---------------
>
> ## Other
>
> * Clarify zero shot accuracy: Yes, the text encoder encodes each of the 1000 classes represented as text into a class embedding. Then, given an image embedding, it is a 1-NN retrieval across the 1000 ImageNet class embeddings. We added more information about this in Section 6.2
> * Tables with hyperparameters: Appendix C, D, E and G now reference the entire training configuration as Python Code.

---

> ### Author Response · Authors · 2023-03-23
> **Response 2**
>
> In general, architectural changes predate theory, and it takes follow-up work from the community to understand why certain components help generalize better (eg, Residual connections, BatchNorm, LayerNorm itself.). Section 7 provides some insights showing that all sub-components of Dual PatchNorm are necessary (eg, centering, scaling, learnable parameters). However, we attempt to provide more insights in the latest revision.
>
> ----
> ## Visualize scale of gradients
> We provide some insight by visualizing the scale of gradient norms. We had this in Appendix D of the previous version, but we moved it to the main section with more details.
>
> We report per-layer gradient norms with and without DPN on B/16. Fig. 3 (Left) plots the mean gradient norm of the last 1000 training steps as a function of depth with and without DPN. Interestingly, the gradient norm of the base ViT patch embedding (black) is disproportionately large compared to the other layers. Applying DPN (red), on the other hand, scales down the gradient norm of the embedding layer. Fig. 3 (Right) additionally shows that the gradient norm of the embedding layer is reduced not only before convergence but also throughout the course of training. This property is consistent across ViT architectures of different sizes (Appendix H).
>
> We present this as an interesting observation, without making strong claims since it is not quite straightforward to causally relate this to better generalization.
>
> ----
>
> ## How did we arrive at this?
>
> We arrived at the Dual PatchNorm solution because of another project that explored adding whitened (decorrelated) patches to ViT. Our initial prototype had a LayerNorm right after the decorrelated patches, to ensure that they are of an appropriate scale. This lead to improvements across multiple benchmarks, suggesting that whitened patches can improve image classification. We later found out via ablations, that just LayerNorm is sufficient at the inputs and adding whitened patches on their own could degrade performance. Our paper highlights the need for rigorous ablations of complicated algorithms to arrive at simpler solutions which can be equally or even more effective.
>
> ----
>
> ## Scale visualizations
>
> We hypothesize that the variance in channels could be because each channel may not be equally important for classification, depending on the data distribution. First, across all architectures, we notice that the first scale parameter of DPN upweights G, R and B in that order.
>
> For each vanilla ViT architecture without DPN, we trained 3 models in the same setup, but take as input, one of R, G and B channels (instead of all 3 RGB channels). For ViT-B models, the validation accuracy ordering is G, R and B, and thus correlates with the scale parameter. For the small models, the validation accuracy ordering is R, G and B. However, the train loss ordering is G, R and B. indicating that the augmentation hyperparameters may be heavily tuned for the RGB + Small baselines. As for the corners of the patch, we leave this as a curious observation.
>
>
> Let us know if you need further clarifications!

---

> > ### Comment · Reviewer_tHeX · 2023-03-24
> > **Thank you for the updates**
> >
> > I want to thank the authors for adding experiments on transfer learning and for a another task besides classification, i.e. semantic segmentation. This is an empirical study is now stronger and more complete, I think. The new experiments validate the gains observed on ImageNet from this simple change.

---

### Review · Reviewer_oWPk · 2023-03-08

**Summary Of Contributions:**

The authors propose Dual Patch Norm (DPN). A light weight extension to common transformer based vision models that adds LayerNorm layers before and after patch encoding. The work provides some empirical evidence that DPN improves classification performance on image classification and contrastive learning.

**Audience:**

Yes

**Broader Impact Concerns:**

no concerns

**Claims And Evidence:**

Yes

**Requested Changes:**

Considering the claims of the paper, the work is already in a good state. Addressing my raised concerns would strengthen the work.

**Strengths And Weaknesses:**

The paper provides a light-weigh, easy to implement extension for transformer based vision models showing some performance improvement, complementing the toolbox of techniques to design and train transformer based models. The paper is well written and structured.

May major criticism is that there is no motivation / argument for the specific DPN beside the empirical results. Maybe the authors could emphasize the cases in which they improved performance and lay out some argument / rational why DPN improves performance. Do the authors see any restrictions of their approach, e.g. considering other tasks (NLP)?

---

> ### Author Response · Authors · 2023-03-23
> **Response**
>
> Thanks for the review!
>
> ## Visualize scale of gradients
> ------
>
> We provide some insight by visualizing the scale of gradient norms. We had this in Appendix D of the previous version, but we moved it to the main section with more details.
>
> We report per-layer gradient norms with and without DPN on B/16. Fig. 3 (Left) plots the mean gradient norm of the last 1000 training steps as a function of depth with and without DPN. Interestingly, the gradient norm of the base ViT patch embedding (black) is disproportionately large compared to the other layers. Applying DPN (red), on the other hand, scales down the gradient norm of the embedding layer. Fig. 3 (Right) additionally shows that the gradient norm of the embedding layer is reduced not only before convergence but also throughout the course of training. This property is consistent across ViT architectures of different sizes (Appendix H).
>
> We present this as an interesting observation, without making strong claims since it is not quite straightforward to causally relate this to better generalization.
>
> -----
>
> ## How did we arrive at this?
>
> We arrived at the Dual PatchNorm solution because of another project that explored adding whitened (decorrelated) patches to ViT. Our initial prototype had a LayerNorm right after the decorrelated patches, to ensure that they are of an appropriate scale. This lead to improvements across multiple benchmarks, suggesting that whitened patches can improve image classification. We later found out via ablations, that just LayerNorm is sufficient at the inputs and adding whitened patches on their own could degrade performance. Our paper highlights the need for rigorous ablations of complicated algorithms to arrive at simpler solutions which can be equally or even more effective.
>
> -----
>
> For NLP in particular, the input layer consists of discrete symbols, so one could try adding a layernorm after the embedding layer.
>
> Let us know if you need further clarifications!

---

### Review · Reviewer_pmPK · 2023-03-13

**Summary Of Contributions:**

This paper proposes a dual PatchNorm (DPN) technique to facilitate the effective training of vision Transformers. In specific, two Layer Normalization layers (LayerNorms), before and after the patch embedding layer in Vision Transformers. Results on ImageNet and JFT demonstrate the effectiveness of DPN.

**Audience:**

Yes

**Broader Impact Concerns:**

I have no concerns about the broader impacts and the ethical implications.



**Claims And Evidence:**

No

**Requested Changes:**

See the weaknesses above.

**Strengths And Weaknesses:**

Strengths:
1) DPN is simple and effective.

Weaknesses:

1) Given the simplicity of DPN, I think the authors should conduct extensive experiments to validate its effectiveness, such that the contributions of this paper will be sufficient to be published. However, the following results are absent. Personally, I'll lean toward accepting this paper if at least two of them are provided, and the results are solid.
- Results on top of DeiT (Training data-efficient image transformers & distillation through attention, ICML'21), a widely-used, state-of-the-art pipeline for training ViTs efficiently. Providing the results with DeiT III (ECCV'22) would be more impressive.
- Results on multi-stage ViTs (e.g., Swin or PVT) or ConvNets (e.g., ConvNeXt).
- Results on downstream tasks (e.g., detection and segmentation on COCO).

2) In addition, this paper may lack some necessary insights. I'm pretty curious about why DPN yields signification improvements. It seems that the current paper does not answer this question. The authors may consider visualizing the scale of gradients in ViTs.

3) In Tables 2 and 3, the gains of DPN are not significant for relatively large models.

---

> ### Author Response · Authors · 2023-03-23
> **Response**
>
> Thanks for the review! We updated a revision with all changes marked in red.
>
> -------------------------------------------
> ## Results on downstream tasks
> We added results on downstream transfer on the VTAB benchmark and Semantic Segmentation on ADE20K. VTAB results are in Section 6.1 + Table 3 and the semantic segmentation results are in Section 6.2 + Table 5. Here are more details.
>
> ### Finetuning on VTAB
> -----------------
>
> We finetune ImageNet-pretrained B/16 and B/32 with and without DPN on the Visual Task Adaption benchmark (VTAB). VTAB consists of 19 datasets: 7 Natural , 4 Specialized  and 8 Structured . Natural consist of datasets with natural images captured with standard cameras, Specialized has images captured with specialized equipment and \taskStructured require scene comprehension. We use the VTAB training protocol which defines a standard train split of 800 examples and a validation split of 200 examples per dataset. We perform a lightweight sweep across 3 learning rates on each dataset and use the mean validation accuracy across 3 seeds to pick the best model. Appendix E references the standard VTAB finetuning configuration. We then report the corresponding mean test score across 3 seeds in Table 3. In Table 3 accuracies within $95 \%$ confidence interval are not bolded.}
>
> On Natural, which has datasets closest to the source dataset ImageNet, B/32 and B/16 with DPN significantly outperform the baseline on 7 out of 7 and 6 out of 7 datasets respectively. Sun397 is the only dataset where applying DPN performs worse. In Appendix F, we additionally show that DPN helps when B/16 is trained from scratch on Sun397. Applying DPN on Structured improves accuracy on 4 out of 8 datasets and remains neutral on 2 on both B/16 and B/32. On Specialized, DPN improves on 1 out of 4 datasets, and is neutral on 2. To conclude, DPN offers the biggest improvements, when finetuned on Natural. On Structured and Specialized, DPN is a lightweight alternative, that can help or at least not hurt on a majority of datasets.
>
> ### Finetuning on Semantic Segmentation
> -----------------
> We finetune ImageNet-pretrained B/16 with and without DPN on the ADE-20K $512 \times 512$ Semantic Segmentation task. Following [Strudel2021 et. al], a single dense layer maps the ViT features into per-patch output logits. A bilinear upsampling layer then transforms the output distribution into the final high resolution $512 \times 512$ semantic segmentation output. We finetune the entire ViT backbone with standard per-pixel cross-entropy loss. Appendix G specifies the full set of finetuning hyperparameters. Table 5 reports the mean mIOU across 10 random seeds and on different fractions of training data. The improvement in IoU is consistent across all setups.
>
> ---------------
>
> ## Results on DeiT
>
> We added results on DeiT-B and DeiT-S to Table 1. Further, we added ViT-AugReg results fine tuned on higher resolution 384x384. High Resolution finetuning is pretty lightweight, we finetune for just 5000 steps with a batch-size of 512 as compared to pretraining (~90000 steps with a batch-size of 4096).
>
> On top of DeiT-S and DeiT-B, DPN provides an improvement of 0.3 and 0.2 respectively. Further, we finetune B/16 and B/32 models with and without DPN on high resolution ImageNet ($384 \times 384$) for 5000 steps with a batch-size of 512 (See Appendix D for the full hyperparameter setting). Applying DPN improves high-res, finetuned B/16 and B/32 by 0.6 and 1.0 respectively.
>
> ---------------
>
> ## Results on multi-stage ViTs
>
> Our claim is that DPN is a lightweight alternative that improves over isotropic, vanilla VIT’s, without any convs or inductive biases from convolutional architectures. Multi-stage ViTs that incorporate inductive biases from convolutional networks are out of scope for this paper, but could be an interesting avenue of future study.
>
> -----------------
>
> ## Gains not significant for larger models
> We dug a bit into this and found that we pretrain models on JFT with a sigmoid loss but we finetune them on ImageNet with a softmax loss. On fixing this, that is when we use sigmoid loss during both pretraining and finetuning, the ImageNet accuracies of both the baseline and DPN improve but DPN improves more. Previously on 4 out of 4 cases, on L/16, (the largest model we trained), DPN matched the baselines. After fixing the loss function during finetuning, DPN now outperforms the baseline in 3 out of 4 cases. Please see Table 2 for new results.
>
> -----------------

---

> ### Author Response · Authors · 2023-03-23
> **Response 2**
>
> ------
> ## Visualize scale of gradient norm
>
> Thanks for the suggestions! We had this in Appendix D of the previous version, but we moved it to the main section with more details.
>
> We report per-layer gradient norms with and without DPN on B/16. Fig. 3 (Left) plots the mean gradient norm of the last 1000 training steps as a function of depth with and without DPN. Interestingly, the gradient norm of the base ViT patch embedding (black) is disproportionately large compared to the other layers. Applying DPN (red), on the other hand, scales down the gradient norm of the embedding layer. Fig. 3 (Right) additionally shows that the gradient norm of the embedding layer is reduced not only before convergence but also throughout the course of training. This property is consistent across ViT architectures of different sizes (Appendix H).
>
> We present this as an interesting observation, without making strong claims since it is not quite straightforward to causally relate this to better generalization.
>
> -----
>
> ## Sidenote
>
> We arrived at the Dual PatchNorm solution because of another project that explored adding whitened (decorrelated) patches to ViT. Our initial prototype had a LayerNorm right after the decorrelated patches, to ensure that they are of an appropriate scale. This lead to improvements across multiple benchmarks, suggesting that whitened patches can improve image classification. We later found out via ablations, that just LayerNorm is sufficient at the inputs and adding whitened patches on their own could degrade performance. Our paper highlights the need for rigorous ablations of complicated algorithms to arrive at simpler solutions which can be equally or even more effective.
>
> -----
>
> Let us know if you need further clarifications!

---

> > ### Author Response · Authors · 2023-03-27
> > **More results on Semantic Segmentation (L16 pretrained on JFT)**
> >
> > Semantic Segmentation results on L/16 (pretrained on JFT) took a bit longer, but we show some more results, that Dual PatchNorm can help even in this regime, where vanilla ViTs have excelled. DPN achieves $53.3 \pm 0.06$ IoU vs the baseline that achieves $52.9 \pm 0.09$ IoU.

---

### Decision · Action_Editors · 2023-05-06

**Recommendation:** Accept as is

**Comment:**

This paper proposes a dual PatchNorm to train vision transformers. It is a simple but effective method, i.e., inserting layer normalization layers before and after the patch embedding layer.  Extensive experiments are conducted to investigate the method's effectiveness.

In the rebuttal phase, the authors add more explanations and investigate the dual PatchNorm on more tasks. All the reviewers satisfy the responses and recommend accepting this manuscript.  The claim in this paper is supported by convincing experiments and many individuals will be interested in it.

Overall, I recommend accepting this paper.

**Audience:**

Yes.

**Claims And Evidence:**

Yes.